# Effect of Neutron Radiation on ^10^BPA-Loaded Melanoma Spheroids and Melanocytes

**DOI:** 10.3390/cells14030232

**Published:** 2025-02-06

**Authors:** Monika Szczepanek, Michał Silarski, Agnieszka Panek, Anna Telk, Katarzyna Dziedzic-Kocurek, Gabriele Parisi, Saverio Altieri, Ewa Ł. Stępień

**Affiliations:** 1Doctoral School of Exact and Natural Sciences, Jagiellonian University, 30-348 Kraków, Poland; monika.szczepanek@ifj.edu.pl; 2Faculty of Physics, Astronomy and Applied Computer Science, M. Smoluchowski Institute of Physics, Department of Medical Physics, Jagiellonian University, 30-348 Kraków, Poland; k.dziedzic-kocurek@uj.edu.pl; 3Faculty of Physics, Astronomy and Applied Computer Science, M. Smoluchowski Institute of Physics, Department of Experimental Particle Physics and Applications, Jagiellonian University, 30-348 Kraków, Poland; 4Department of Biological Physics and Nanospectroscopy, Institute of Nuclear Physics, 31-342 Kraków, Poland; agnieszka.panek@ifj.edu.pl; 5Department of Analytical Chemistry, Faculty of Chemistry, Jagiellonian University, 30-387 Kraków, Poland; anna.telk@uj.edu.pl; 6Department of Physics, University of Pavia, 27100 Pavia, Italysaverio.altieri@unipv.it (S.A.); 7Nuclear Physics National Institute (INFN), 27100 Pavia, Italy; 8Centre for Theranostics, Jagiellonian University, Kopernika 40, 31-501 Kraków, Poland

**Keywords:** boron neutron capture therapy, melanoma, melanocytes, spheroids, boronophenylalanie, DNA damage, ICP-MS, thermal neutrons

## Abstract

Melanoma is an aggressive disease that arises from mutations in the cells that produce the pigment melanin, melanocytes. Melanoma is characterized by a high mortality rate, due to avoidance of applied therapies and metastasis to other organs. The peculiar features of boron neutron capture therapy (BNCT), particularly its cell-level selectivity, make BNCT a promising modality for melanoma treatment. However, appropriate cellular models should be used to study new therapies or improve the efficacy of existing therapies. Spheroids, which have been used for years for in vitro studies of the efficacy of anti-cancer therapies, have many characteristics shared with tumors through which they can increase the accuracy of the cellular response compared to 2D culture in vitro studies and reduce the use of animals for research in the future. To the best of our knowledge, when we started researching the use of spheroids in BNCT in vitro, there was no publication showing such use. Our study aimed to evaluate the efficacy of a 3D cellular model (spheroids) for testing BNCT on melanoma cells. We assessed boronophenylalanine (^10^BPA) uptake using inductively coupled plasma mass spectrometry in both spheroids and 2D cultures of melanoma and melanocytes. DNA damage, Ki67 protein expression, and spheroid growth were analyzed. The experimental groups included: (1) IR_B (neutron flux + 50 µg ^10^B/mL), (2) IR (neutron flux alone), (3) C_B (no irradiation, 50 µg ^10^B/mL), and (4) C (no irradiation and no treatment with boron). The total absorbed doses were estimated to be 2.1–3.1 Gy for IR_B cells and spheroids as well as 8.3–9.4 Gy for IR_B spheroids, while estimated doses for IR cells were 0.5–1.9 Gy. The results indicated that IR_B spheroids might exhibit a reduced diameter. Melanoma cells in the 3D model showed that their DNA damage levels may be higher than those in the 2D model. Moreover, the Ki67 assay revealed differences in the expression of this marker between irradiated melanoma cell lines. In conclusion, preincubation with ^10^BPA enhances BNCT efficacy, leading to cell growth inhibition and increased DNA fragmentation. Differences in DNA damage between 2D and 3D models may be due to dissimilarities in cell metabolism caused by a changed cell architecture.

## 1. Introduction

Boron neutron capture therapy (BNCT) is a two-step treatment involving the administration of a compound containing the ^10^B isotope to the patient. One of the two boron carriers approved for clinical use is ^10^BPA (4-borono-l-phenylalanine). The increased metabolic activity of cancer cells is exploited in BNCT, also associated with increased expression of transporters from the SLC (LAT) protein family, which results in a higher concentration of the ^10^B compound in cancer cells compared to normal cells [1,2]. For effective therapy, the boron concentration ratio between cancer and normal cells should be at least 3:1 [3,4,5]. The affected area of the patient’s body is then irradiated with a field of thermal or epithermal neutrons, depending on the tumor location. In cancer cells loaded with boron, due to its high cross-section for the thermal neutron capture process (3835 barns at 25 meV of neutron energy), the ^10^B isotope turns into an unstable, short-living isotope ^11^B, which decays into short-range highly energetic particles: an α particle and a lithium nucleus (^7^Li). Their high linear energy transfer (LET) results in high-dose deposition and short tissue penetration of less than 10 µm, which corresponds to the size of a single cell [6]. Thus, only the cells containing ^10^B are damaged, with minimum influence on the surrounding normal cells [3,5,7]. Both of these factors, the appropriate concentration of ^10^B in the tumor cells (20–35 µg ^10^B/g of tissue), and proper neutron field, enable highly selective tumor destruction by BNCT [5,8]. The greatest advantage of this therapy is the selective destruction of tumor cells without requiring precise knowledge of the tumor’s location. Although there has been increasing interest in BNCT recently, mainly through the development of technology to provide neutron sources capable of being installed inside the hospital environment and thus the establishment of new patient centers, there are still several aspects that need to be improved to increase the effectiveness of the therapy [6].

The focus of the presented work is a preclinical evaluation of BNCT for melanoma. For this purpose of in vitro research, an appropriate cellular model is required [9,10,11]. Recently, spheroids have been increasingly used as a model in cancer therapy research, due to their many similarities to solid tumors [12,13,14,15]. These include layered structures with the outermost layer composed of proliferating cells, a middle layer of viable but nonproliferating cells, and a central necrotic core of dead cells. This organization, driven in part by the limited availability of nutrients and oxygen, participated in the creation of a tumor-like microenvironment characterized by lower pH, hypoxia, and the accumulation of metabolic waste [16,17]. Additionally, the cell–cell contact, cell signaling, and extracellular matrix deposition observed in both tumors and 3D cultures form some protection barriers against drug penetration and radiotherapy. Both structures show similarities in the growth kinetics or expression patterns of certain genes, mainly involving proliferation, migration, and invasion. The similarities between spheroids and solid tumors make spheroids a promising in vitro model that could reflect tumor response to therapy [16,17,18]. One promising application of BNCT is in the treatment of melanoma. It is a very aggressive form of skin cancer originating in melanocytes, the melanin-producing cells in the epidermis. The high mortality rate of melanoma patients is associated with late diagnosis, poor prognosis, and limited treatment efficacy [19]. BNCT offers a potential opportunity to prolong survival and improve the quality of life in the case of this disease [6].

The purpose of our study was to evaluate melanoma spheroids as a model for in vitro testing of the BNCT effect. We investigated melanoma cells in two models (2D and 3D cell culture) as well as melanocytes as a control (2D) to assess DNA damage after BNCT and neutron radiation alone. In addition, we evaluated the growth inhibition of melanoma spheroids (measuring changes in diameter and Ki67 protein levels), and DNA damage after BNCT at two different radiation doses, with and without the presence of ^10^B in the cells.

## 2. Materials and Methods

### 2.1. Cell Cultures

Commercially available human melanoma cell lines FM55p and WM266-4 obtained from the ESTDAB Melanoma Cell Bank (Tübingen, Germany) were used. The cells were cultured in a RPMI 1640 medium (Cat. No. 21875091 Gibco™ Paisley, UK) supplemented with 10% Fetal Bovine Serum (Cat. No. 10500064 Gibco™ Paisley, UK), 2 mM L-Glutamine (Cat. No. 25030081 Gibco™, Paisley, UK), 100 U/mL Penicillin, and 100 µg/mL Streptomycin (Cat. No. 15140122 Gibco™). The melanocyte cell line HEMa-LP was cultured in M254 medium (Cat. No. M254500 Gibco™ Paisley, UK) with Human Melanocyte Growth Supplement-2 (Cat. No. S0165 Gibco™ Paisley, UK), 100 U/mL Penicillin, and 100 µg/mL Streptomycin (Cat. No. 15140122 Gibco™). The cells were seeded into a T75 cm^2^ dish and cultured at 37 °C and a 5% CO2 atmosphere. The culture medium was replaced every second day.

Melanoma 3D spheroids were obtained in a 96-well U-shaped low adhesive plate (Cat. No. 34896 3D^TM^ Cell Floater Plate SPL Life Science, Seoul, Republic of Korea) after seeding 2000 cells per well. Spheroids were incubated at 37 °C in a humidified atmosphere of 5% CO_2_, and the 75% culture medium was replaced every second day.

### 2.2. BPA-Fructose Solution Preparation

BPA solution was prepared at a concentration of 0.14 M. ^10^B-enriched p-boronophenylalanine (Cat. No. 00905018, Interpharma Praha, Prague, Czechia) was mixed in water with a 10% molar excess of fructose [20]. The pH of the solution was changed to alkaline with NaOH to allow easier dissolution of BPA crystals. Then, the solution was stirred up until the BPA was completely dissolved, and then the pH was neutralized with HCl.

### 2.3. Boron Uptake Experiments

The cells in the exponential growth phase and the 7-day-old spheroids were used to determine the uptake of ^10^B. The cells growing in T75 cm^2^ dishes and spheroids in plates were incubated for 2 h, 4 h, 6 h, and 24 h with BPA (50 µg ^10^B/mL). The control samples were incubated with a standard growth medium. After incubation, the medium was removed and the cells were washed three times with PBS w/o Ca^2+^, Mg^2+^ (Cat. No. 10010056 Gibco^TM^ Paisley, UK). Next, the cells were harvested by trypsinization (0.25% trypsin (Trypsin—EDTA, Cat. No. 25200072 Gibco™ Paisley, UK)) and centrifuged at 260× *g* for 10 min (Hermle Z300K, Wehingen, Germany). Subsequently, pellets of cells were resuspended in fresh medium, and the cells were counted with trypan blue dye (Cat. No. T8154, Sigma Aldrich, Taufkirchen, Germany) in the automatic cell counter LUNA II. A single-cell suspension was transferred into 5 mL Eppendorf tubes and centrifuged at 300× *g* for 5 min (Eppendorf 5424R, Eppendorf, Köln, Germany). Regarding the spheroids, they were collected in tubes after incubation, and after centrifugation, they were washed three times with PBS w/o Ca^2+^, Mg^2+^. The pellets of cells (spheroids) were weighed (cell mass was 50–100 mg) and 1 mL of concentrated nitric acid was added to each sample, then they were left for 1.5 h under the fume hood and finally placed in the refrigerator until ICP-MS measurement.

### 2.4. Inductively Coupled Plasma Mass Spectrometry (ICP-MS) Measurements

To confirm boron uptake by cells and exclude any possible contamination, ICP-MS measurements were performed. Samples previously pretreated with nitric acid were transferred into Teflon hermetic vessels and subjected to mineralization supported by microwave radiation (Anton Paar, Graz, Austrian, Multiwave 3000, 4 × XF100, max. 60 atm., p-rate 0.4 bar/s, IR 240 °C). A digest of the samples was diluted 10 times with 1% *v*/*v* nitric acid directly before the analysis by an inductively coupled plasma mass spectrometer (Perkin Elmer Elan DRC-e) with the parameters given in Table 1. The boron concentration was measured as the ^10^B isotope, with the use of a calibrator curve prepared using a Merck ICP-MS Multi Element Standard no. IV (23 Elements) Centipur, Taufkirchen, Germany, where the abundances of boron in the standard were: ^10^B—19.9% and ^11^B—80.1%. Since the cells were incubated with a medium with a monoisotopic compound (^10^B), mathematical corrections were made on the obtained results. After, each sample’s blank signal was checked and an additional standard was used to control the stability of the measurements (boron standard solution of 10 mg/L in water dedicated to ICP-MS, ARISTAR).

### 2.5. Neutron Irradiation. Experimental Design

Cells were irradiated in the thermal column of the Triga Mark II research nuclear reactor of the Applied Nuclear Energy Laboratory (L.E.N.A.) of Pavia University. Samples were placed in the thermal column as shown in Figure 1, 130.95 cm from the center of the reactor core, in an air channel with dimensions 40 × 20 × 100 cm^3^. It is made of graphite blocks which thermalize the fission neutron and contains two bismuth filters to reduce the gamma radiation background coming from the reactor core. Cells were irradiated with thermal neutrons so that the 2D cultures containing boron received a dose of about 2 Gy. In the case of spheroids, the delivered dose was about 2 Gy for the first set of 96-well plates and 6 Gy for the second one. The thermal neutron flux at the sample irradiation position can be determined directly from the reactor power. Thus, the reactor powers for irradiation were determined using Equation (1) [21,22] to achieve the assumed doses with an irradiation time $t_{irr}$ fixed to 600 s. The reactor power, thermal neutron flux, and doses received by cells (with and without boron) are shown in Table 2. Each cell line, containing boron and without it, was irradiated with the same thermal neutron fluence.

The total dose $D_{Tot}$ administered to the cells during the time interval $t_{irr}$ with the reactor power at the level of *P* (in kW) was calculated based on the following parametrization given in Ref. [22]:(1)$$D_{Tot}=\left(R_{bg}+C_{B}R_{B}\right)t_{irr}\frac{P}{P_{max}\;}$$
where $R_{bg}$ is the background dose rate equal to the sum of contributions due to the following: ^14^N(n,p)^14^C reaction, neutron scattering on hydrogen, and γ-rays at the irradiation position (including the 2.2 MeV photons from the ^1^H(n,γ)^2^H reaction). The dose rate associated with boron was estimated by determination of the ^10^B (n,α)^7^Li reaction contribution assuming 1 ppm of ^10^B multiplied by the concentration, C_B, the latter determined for each cell line by ICP-MS measurements before the irradiation. Both $R_{bg}$ and $R_{B}$ were estimated assuming that the reactor was working at the maximum power (250 kW).

The $R_{bg}$ and $R_{B}$ dose rates were determined using Monte Carlo simulations with the PHITS v3.341 [23] and MCNPv6.2 [24] codes. Simulations were carried out in two steps. The MCNP code was used to determine the neutron and γ-ray background fluxes at the irradiation position. Both the geometry and the models of particle interaction with the reactor material and samples were optimized based on the previously performed dosimetric measurements in the thermal column [22]. The characterization of the neutron field and the accompanying γ flux were performed using both Monte Carlo methods (MCNP5) and activation measurements (set of Au, In, Mn, Cu, W foils) for neutron energies up to several MeV. Alanine dosimeters were used to measure doses from gamma quanta. Details of the simulation tuning and the dosimetric measurements can be found in [22]. With MCNP, we have determined the energy distribution and flux of neutrons and γ-rays passing through a cube of 20 cm sides surrounding the irradiated samples. These data were used as a source for the PHITS simulations performed to determine the doses delivered to the irradiated cells. We have implemented detailed geometry of the T25 cell culture flask for the 2D cultures and the 96-well plate for spheroids assuming they are made of 1 mm thick polystyrene (the elemental composition was taken from [25]). Based on the known volume of the samples (1.5 cm^3^) and the flask area (25 cm^2^), we have implemented an 80 μm thick layer of medium and 10 μm thick 2D cell cultures. In the case of spheroids, the 1 mm cell spheres immersed in 0.2 cm^3^ of medium were simulated. The elemental composition of the medium was taken from Ref. [21] while cells were simulated as the average adult male healthy soft tissue according to the ICRU-44 recommendations [26]. The dose rates were estimated for each of the cell flasks and each of the spheroids separately, assuming 1 ppm of ^10^B concentrations for $R_{B}$. The values are reported in Table 3.

These values were also quite uniform for all the simulated 2D cell cultures and spheroids (the standard deviation of the estimated dose rates did not exceed 1%). The mean values of $R_{bg}$ and $R_{B}$ obtained for cells cultured in 2D and spheroids are presented in Table 3 together with the contributions caused by neutron interactions with nitrogen and hydrogen, and by γ-rays.

Using the calculated values of $R_{bg}$ and $R_{B}$ we estimated doses delivered to the irradiated cell cultures according to Equation (1). The obtained values are gathered in Table 2 alongside the measured boron concentrations, used reactor powers, and neutron fluxes. Since the precise elemental content of the studied cell cultures is not known, we have checked the influence of the assumed mass elemental ratios on the estimated doses by repeating the calculations for ICRU-44 human skin [26]. Significant changes in the dose values were found for the nitrogen only, and thus, we reported them as systematic uncertainty of the calculations. As one can see, the dose values for spheroids are generally higher than those for the 2D cultures. In the following sections, for simplicity, all cells and spheroids irradiated with neutrons after incubation with or without BPA are referred to as the lower dose group (2 Gy of reference dose) and the higher dose group (6 Gy of reference dose).

Cells and spheroids were divided into the following groups: (1) IR_B: cells incubated with ^10^BPA for 4 h before neutron irradiation; (2) IR: cells without ^10^BPA and irradiated with neutrons; (3) C_B: cells incubated with ^10^BPA for 4 h and not exposed to neutrons; (4) C: cells without ^10^BPA and not irradiated with neutrons. Cells were grown and irradiated in T25 cm^2^ flasks (2D culture) and 96-well plate (spheroids). All experiments were conducted in triplicate. Biological assays were carried out 1 h, 24 h, and 48 h after neutron irradiation and they included the following: a DNA damage assay, spheroids proliferation assay, as well as microscopic imaging (Figure 2).

### 2.6. DNA Damage Level—Comet Assay

Cells and spheroids were collected as described above. The pellet of the cells was resuspended in a freezing medium and stored at −80 °C until the test. To avoid crystal formation, the melanoma cells were frozen in the RPMI medium with 20% FBS and 10% dimethyl sulfoxide, DMSO (Cat. No. D12345, Invitrogen™, Grand Island, NY, USA), while melanocytes were frozen in Medium 254 with 20% FBS and 10% DMSO.

DNA damages were tested by the standardized alkaline comet assay, as described by Panek A. et al. [27] with minor modifications. Cells were thawed in a water bath (37 °C), immediately transferred into 4 mL of standard growth medium, and centrifuged in a chilled centrifuge at 750× *g* (Hermle Z300K) for 3 min. Next, cells were mixed with 0.5% low melting agarose gel in a ratio of 1:3 and transferred onto a pre-coated basic slide with 1% agarose gel. Cells were lysed by incubating for 1 h at 4 °C in a pre-chilled lysis buffer (2.5 M NaCl, 100 mM EDTA, 10 mM TRIS, 175 mM NaOH, pH = 10 mixed with 10% DMSO and 1% Triton-X) immediately before use. Next, slides were washed for 3 min in distilled water and transferred into the chilled electrophoresis chamber, and they were incubated for 20 min at 4 °C in pre-chilled electrophoresis buffer (0.3 M NaOH, 1 mM EDTA in chilled distilled water, pH > 13). Alkaline electrophoresis was performed under the following parameters: 4 °C, 28 V, and 300 mA for 30 min. After the electrophoresis DNA was neutralized by washing the slides three times for 5 min each in tris buffer, pH = 7.5. Finally, DNA was stained with ethidium bromide (4 µg/mL) and visualized using the epifluorescence microscope Olympus BX-50 (100 W mercuric lamp, excitation filter 515–560 nm, barrier filter from 590 nm; Olympus, Tokyo, Japan) connected to a CCD camera (Pulnix, Kinetic, Liverpool, UK). The Komet 3.0 program (Kinetic Imaging Company, Liverpool, UK) was used to analyze the comet images. DNA damage was quantified using the DNA tail parameter (percentage of DNA in the comet tail), where changes in its distribution are considered as a sensitive indicator of DNA damage. For each sample, 100–200 cells were counted in three replicates.

### 2.7. Spheroids Growth Analysis

An average of 43 spheroids images were taken just before neutron irradiation and also 1 h, 24 h, and 48 h after irradiation. The spheroids’ size (diameter) and shape (circularity) were analyzed with the ImageJ software (v 1.52p) using a customized macro. The spheroid diameter was normalized to the diameter measured before irradiation.

### 2.8. Proliferation Assay

At 1 h, 24 h, and 48 h after irradiation, six spheroids were collected into a 1.5 mL Eppendorf tube. Spheroids were dissociated as described above. The pellet of cells was fixed in ice-cold 70% ethanol and stored at −20 °C until analysis. Fixed cells were prepared for cytometer analysis according to the manufacturer’s protocol. Briefly, cells were washed with PBS with 1% FBS and centrifuged at 500× *g* (Hermle Z300K) for 10 min twice and resuspended to a concentration of 1 × 10^7^/mL. Next, 100 µL of cell suspension was transferred into a 5 mL tube and 20 µL of anti-Ki67 antibody (Cat. No. 556026, BD Biosciences, Warszawa, Poland) was added, mixed gently, and incubated for 30 min at room temperature in the dark. Then, cells were washed with PBS with 1% FBS and centrifuged at 500× *g* (Hermle Z300K) for 5 min and resuspended in PBS with 1% FBS and PI was added. Cells were analyzed with a Cell Stream flow cytometer (Luminex, Darmstadt, Germany). The fluorescence signal was collected in channel C3 (for Ki67) for excitation at 488 nm and emission at 528/46 nm, and in channel C6 (for PI) for excitation at 488 nm and emission at 702/87 nm. Analysis was performed using the Cell Stream Analysis 1.2.272 software.

### 2.9. Statistical Analysis Applied to the Obtained Data

The statistical significance of compared data samples was quantified using the two-sample *t*-test at the significance level of 95% with the alternative hypothesis assuming significant differences between means of each sample (not assuming the equality of variances). In the cases of DNA damage scoring, spheroid shape parameters, and the proliferation assay Ki67 expression level, the uncertainties were estimated as the variance of the mean with the following formula:$$∆=\sqrt{\frac{1}{n(n-1)}\sum_{i=1}^{n}{\left(\bar{t}-t_{i}\right)}^{2}}\;,$$
where *n* is the size of the sample, $t_{i}$ are the measured values, and $\bar{t}$ represents their mean value. Additionally, the uncertainties of the estimated doses delivered to the cells were calculated using the error propagation law taking into account the uncertainties of all the quantities used in Equation (1). The dominant contribution to these uncertainties appears to be the boron concentration listed in Table 2.

## 3. Results

### 3.1. Boron Uptake Study

The boron concentration in cancer and normal cells is a crucial factor influencing the effectiveness of BNCT. We tested different incubation times of cells with BPA to determine the optimal time at which maximum boron concentration in the cells is achieved for subsequent experiments. Figure 3 presents changes in boron concentrations in cells as a function of time for melanocytes and melanoma cell lines in 2D and 3D cell cultures. Simultaneously, a viability test was conducted to confirm that no significant cell death was induced. The viability of cells was always higher than 95% for all three tested cell lines and for each incubation time, which confirmed that BPA was not toxic to cells. These results are presented in Appendix A. The results shown in Figure 3 confirmed that BPA uptake depends on the cell line and type of culture (2D vs. 3D). The highest boron concentrations in cells were observed at 4 h and 6 h of incubation. Among the 2D models, melanocytes exhibited the highest boron concentrations (0.24 ± 0.02 µg ^10^B/10^6^ cells at 4 h incubation), which was significantly different from those of both metastatic melanoma cells (*p* = 0.0130) and primary melanoma cells (*p* = 0.0002) with the lowest boron concentration (0.06 ± 0.02 µg ^10^B/10^6^ cells). Figure 3 also shows significantly higher boron uptake for melanoma spheroids compared to the same cell line in 2D (*p* = 0.03 for WM266-4 and *p* = 0.02 for FM55p). However, there is no significant difference in boron concentrations between melanocytes and WM266-4 spheroids at 4 h of incubation.

### 3.2. DNA Damage

Comet assay results present a wide spectrum of DNA damage such as single-strand breaks (SSB), double-strand breaks (DSB), alkali-labile (AP) sites, and oxidative base lesions induced by irradiation. For the normal cell line, HEMa-LP, the level of DNA damage was significantly higher in IR_B cells than in IR or control cells 1 h after irradiation. The level of damage in the irradiated samples was then equated to that of the control (Figure 4c). Similarly, for the FM55p 2D cell line, the DNA damage level was significantly higher in IR_B cells than in IR and control cells. However, this damage was maintained even 24 h after irradiation (Figure 4a). In contrast to normal and primary cell lines, no differences in the level of DNA damage were observed for the metastatic cell line (Figure 4b). Comparing the primary cell line in 2D and 3D cultures at a 2 Gy dose, a similar effect was observed. Higher DNA damage was noted in IR_B cells than in IR cells remaining at the same level over time; however, it was greater for spheroids. DNA damage for IR_B spheroids at a 6 Gy dose was higher than for those at a 2 Gy dose, and it decreased over time. In contrast to 2D, in the case of the 3D WM266-4 cell line, DNA damage was observed 1 h after irradiation and was significantly higher in IR_B than in IR spheroids (Figure 4b). Subsequently, the level of DNA damage decreased to the control level. Similarly, for spheroids irradiated with a dose of 6 Gy, the highest level of damage was noted 1 h after irradiation, which was significantly higher than for IR spheroids. Comparable to the FM55p spheroids, the WM266-4 cell line showed a higher level of DNA damage at a dose of 6 Gy compared to 2 Gy.

It is also worth noting that the DNA damage of the FM55p (2D) cell line is at a higher level than that of the WM266-4 (2D) cell line, despite the fact that for the primary cell line, the boron level was below the value required for an effective BNCT reaction (Figure 4d). In contrast, for the 3D culture, the higher boron level measured for the FM55p cell line correlates with a higher level of unrepaired DNA, especially 1 h after irradiation. An increase in the level of unrepaired DNA is also visible for the spheroids of both lines with the same ^10^B concentration irradiated with the higher dose. However, the results shown in Figure 4e may suggest that effective BNCT treatment for these cell lines can be performed for boron concentrations equal to at least 50 ppm.

### 3.3. Spheroids Growth Analysis

The spheroid growth response to irradiation differed between the FM55p and WM266-4 cell lines. As seen in Figure 5a IR_B FM55p, the irradiated spheroids were characterized by growth inhibition compared to the IR and C spheroids already at the dose of 2 Gy. However, for the IR_B spheroids, a diameter increase was observed between 24 and 48 h after irradiation, and it was smaller for the IR_B spheroids irradiated with 6 Gy dose than for those irradiated with the 2 Gy dose. On the other hand, the IR_B WM266-4 spheroids exhibited growth patterns similar to the IR spheroids and the control group in the first 24 h after irradiation. The IR_B spheroids’ growth inhibition was observed after this time, which was more pronounced for the 6 Gy dose case than for the 2 Gy one, similar to the FM55p cell line. The increase in spheroid diameter for the IR spheroids was comparable between both cell lines and similar to controls.

Spheroid shape analysis, indicated by the circularity parameter, did not reveal any trends suggesting an effect of neutron radiation on the shape of FM55p spheroids (Figure 5b). The sole noticeable observation was the increase in circularity among spheroids incubated with BPA. For spheroids not incubated with BPA, nearly a constant level of circularity was observed (Figure 5b). Figure 5c presents a panel of microscopic images of spheroids taken before irradiation and the incubation of BPA (0 h) and 1 h, 24 h, and 48 h after irradiation, along with the corresponding images of spheroids from the control group. In the shape analysis of the WM266-4 spheroids, the only visible change was the increase in circularity values for the irradiated samples (0.87–0.9) with respect to the control spheroids (0.81–0.84) as shown in Figure 6b. A significant decrease in circularity was observed at 1 h after irradiation for the IR_B spheroids exposed to a dose of 6 Gy (from 0.90 to 0.85, *p* < 0.0001). Additionally, a slight decrease in circularity was noted for the IR_B spheroids irradiated with a dose of 2 Gy (from 0.90 to 0.87) and for those exposed to 6 Gy (IR) (from 0.90 to 0.88) at the same time point (for all differences, *p* < 0.0001) (Figure 6b). Figure 6c shows example images of the IR_B and IR spheroids, along with controls (C and C_B). These images illustrate the changes in shape and size of the spheroids, as presented in Figure 6a,b.

### 3.4. Proliferation Assay

In the proliferation assay, the nuclear Ki67 protein was used to investigate the effect of irradiation on the division capability of melanoma cells, both after BPA incubation and without it.

For the primary melanoma cell line FM55p, Ki67 protein levels were the highest at 1 h after irradiation and then decreased for the following hours for the IR_B spheroids at both doses. A similar trend was observed for the IR spheroids (Figure 7a). Additionally, there were no significant differences in Ki67 protein levels between the IR_B spheroids and the IR spheroids at any of the tested time points for both doses. The results were different for the WM266-4 cell line spheroids (Figure 7b). The level of Ki67 proteins decreased at each time after neutron irradiation for the IR_B spheroids irradiated with both doses, 2 and 6 Gy. Moreover, the Ki67 protein level was noticeably different between the IR_B spheroids irradiated with different doses, except in the first hour after radiation. On the other hand, the Ki67 protein level was comparable over time for the IR spheroids, dose-independently. More importantly, for the IR_B spheroids irradiated with a dose of 6 Gy, the level of this protein was significantly lower than for the IR spheroids (except 1 h after irradiation), which was not observed for the lower dose of 2 Gy (Figure 7b).

## 4. Discussion

The 3D cellular models, including spheroids and tumor spheres, are often used in research for finding new biomarkers, or anti-cancer therapies [10,11,29,30]. Spheroids produced from melanoma cells constitute a well-described in vitro research model [9,29,31,32,33,34].

Melanoma cell lines exhibit varying uptake of BPA based on their metabolism, as previously demonstrated by M. Carpano et al. [20]. This study not only showed the biodistribution of ^10^B in different organs but also showed the pharmacodynamics of BPA in melanoma tumors. Therefore, testing boron carriers on in vitro models that best reflect the nature of the tumor is the optimal strategy for pre-clinical BNCT studies. Additionally, it is crucial to measure the concentration of ^10^B to assess the pharmacokinetics of BPA in the tested cell lines [20,35,36,37,38]. In our study, we confirmed that boron concentration in melanoma is dependent on the duration of incubation with BPA, reaching its maximum after 4 h. These changes in kinetics were significant for the 3D culture.

Our research carried out on 2D cell cultures showed the lowest ^10^B concentration in the primary, pigmented FM55p cell line (0.057 ± 0.015 µg ^10^B/10^6^ cells), followed by the metastatic, non-pigmented WM266-4 cell line (0.156 ± 0.002 µg ^10^B/10^6^ cells), while the highest ^10^B concentration was demonstrated by the normal, pigmented HEMa-LP cell line (0.238 ± 0.017 µg ^10^B/10^6^ cells). Opposite results obtained by Faião-Flores et al. show a higher ^10^B concentration in murine melanoma cells (B16F10) than in human melanocytes [39]. However, the melanoma cell line used in their study also had higher levels of melanin compared to melanocytes. These findings support the theory that cells containing melanin have a greater capacity to uptake BPA. Moreover, Coderre et al. examined BPA uptake by differently pigmented melanoma cells, proving that amelanotic melanoma cells were characterized by lower ^10^B concentration than melanin-containing cells [40]. In contrast to these findings, our results indicate that amelanotic melanoma cells had a higher ^10^B concentration than the pigmented melanoma cell line.

Wittig et al., in their publication, tested different temperatures and medium compositions on BPA uptake and efflux [41]. These authors supported the hypothesis about BPA transportation by the L-system for neutral amino acids and provided evidence that BPA taken up by cells is not metabolized or used as a substrate in melanogenesis. These results were not consistent with the theory of greater BPA uptake by pigmented cells and agree with our results. A similar experiment was conducted by Carpano et al., confirming the cell line- and time-dependent uptake of BPA by melanoma cells, and obtaining the highest concentration of ^10^B after 4 h of incubation [20]. They used three different melanoma cell lines that achieved the following ^10^B concentrations: Mel-J—pigmented, metastatic (0.119 ± 0.016 µg B/10^6^ cells), followed by A375—amelanotic, metastatic (0.109 ± 0.007 µg B/10^6^ cells), and the lowest concentration for the M8—amelanotic, primary (0.035 ± 0.007 µg B/10^6^ cells) cell line [20]. For melanoma spheroids, the results for ^10^B concentration were opposite to those of the FM55p and WM266-4 cell lines in 2D culture. The highest ^10^B content was observed in FM55p spheroids at 0.70 ± 0.15 µg B/10^6^ cells at 4 h of incubation, which was 12 times higher than that obtained in 2D culture. In spheroids of the WM266-4 cell line, ^10^B concentration was 0.25 ± 0.03 µg B/10^6^ cells at 4 h of incubation (1.6 times higher than for 2D culture). We would assume that metabolic activity and LAT1 transporter expression contribute to boron uptake and efflux [1,2,42]. Yu et al. also obtained a higher boron concentration in spheroids than in 2D culture after 4 h BPA incubation with human pancreatic cancer cells [43]. They compared two cell lines, Panc-1 and BxPC-3 in the 3D and 2D types of culture and obtained a lower ^10^B concentration for the 3D BxPC-3 cell line than for 2D and the opposite for the Panc-1 cell line. These may suggest that the uptake of ^10^B depends not only on the cell line but also on the type of culture used (2D vs. 3D).

To assess the impact of neutron irradiation with and without ^10^B on melanoma cells in 3D cell culture, we carried out an analysis of the size and shape of spheroids, DNA damage, and cell proliferation. A comet assay was performed for melanoma cells growing in 2D cell culture and for melanocytes (2D). The influence of a 2 Gy dose of neutron irradiation (IR_B) on 2D cell culture was different for the tested cell lines. At 1 h after neutron irradiation, the highest DNA damage level was obtained for melanocytes with 18.3 ± 1.1% of tail DNA, followed by the primary melanoma cell line, FM55p, 16.4 ± 0.5% of tail DNA, and no increase in DNA damage level for metastatic melanoma—the WM266-4 cell line. For the HEMa-LP and FM55p cell lines, DNA damages were significantly higher for IR_B than IR cells. However, for melanocytes, they decreased to control and IR levels 24 h after irradiation while for FM55p, it was still on the same level as measured at 1 h after irradiation. Such an effect on the normal line may indicate an effective DNA repair system. Rodriguez et al. examined DNA damage following BNCT in melanoma cells (Mel-J cell line) by the histone H2AX phosphorylation (γH2AX) analysis [44]. Their results showed high DNA damage in melanoma cells which decreased 24 h after irradiation. A similar observation was made by Chen et al., who studied the impact of BNCT on hepatocellular carcinoma cells [45]. They found the highest incidence of double-strand breaks 4 h post-irradiation, with the damage decreasing over time. Analysis of DNA damage in the FM55p cell line spheroids revealed higher DNA damage (19.9 ± 0.8% of tail DNA) in comparison to 2D cells. Moreover, those damages were always significantly greater for the IR_B (with boron) than IR (without boron) spheroids. Additionally, dose-dependent DNA strand breaks were observed for the IR_B spheroids resulting in higher DNA damage for spheroids irradiated with a dose of 6 Gy (24.4 ± 0.6% tail DNA). For the WM266-4 cell line, DNA damage was observed in the 3D cell culture in contrast to the 2D cell culture. The damage was lower than for primary melanoma spheroids for both doses (14.9 ± 0.9% of tail DNA for 2 Gy and 18.7 ± 0.6% of tail DNA for 6 Gy 1 h after irradiation). On the whole, the results of the comet assay might suggest that spheroids are more susceptible to BNCT than 2D cell cultures. This is in line with the findings published by Yu et al. [43], revealing higher DNA damage in spheroids than in 2D culture for human pancreatic cancer cell lines. However, the variability of the results obtained by the comet assay needs to be further investigated. To this purpose, additional experiments will be carried out to clarify the variability observed and to achieve a robust conclusion.

The BNCT effect, observed as spheroid growth inhibition (diameter) of the FM55p cell line spheroids, combined with high levels of the Ki67 protein, may suggest a cell cycle arrest in the G2/M phase. After 24 h, there was an increase in spheroid diameter (faster for those irradiated with the lower 2Gy dose), and a decrease in Ki67 protein levels. This may indicate cell entry into the G0 phase of divided, intact, or repaired cells. The WM266-4 cell line spheroids were also affected by BNCT. Similarly to FM55p, the WM266-4 spheroids showed dose-dependent growth inhibition after BNCT, demonstrated by both the analysis of spheroid diameters and the Ki67 protein levels. In the IR_B spheroids irradiated with a dose of 2 Gy, growth inhibition (no increase in diameter) is noticeable 24 h after irradiation. In addition, for this dose, the Ki67 protein level was significantly lower than in the control, starting 24 h after irradiation (not before), and did not differ from the protein level of the IR spheroids. In the IR_B spheroids irradiated with a dose of 6 Gy, growth inhibition appeared before the decrease in spheroid diameter. The different results of the proliferation test for melanoma cell lines indicate a different response of the two melanoma cell lines to BNCT treatment. Considering this result together with changes in spheroid diameter, it could mean that BNCT inhibits cell proliferation of the WM266-4 cell line more effectively than FM55p (6Gy dose). These results may also suggest the disintegration of spheroids, which corresponds with the Ki67 protein levels that were significantly lower than in the control and IR spheroids. A substantial reduction in the Ki67 protein level after BNCT in murine melanoma cells was observed by Faião-Flores et al. in comparison to control and neutron radiation [46]. The decrease in spheroid diameter and reduction in the Ki67 protein level may indicate effective damage to the outer layer of proliferating cells caused by BNCT. Yu et al. observed the disassembling of spheroids after BNCT [47]. They showed a high number of double-strand breaks (γH2AX) and the presence of dead cells in the proliferating layer in spheroids suggesting that those cells are more susceptible to BNCT. Cells undergoing irradiation show G2/M cell cycle phase arrest, which was demonstrated on hepatocarcinoma cells [45], glioma stem/progenitor cells, a differentiated human glioma cell line [48], and follicular thyroid carcinoma [49].

For the experiments on the FM55p cell line without ^10^BPA irradiated with a 6 Gy dose, the results for time points at 24 h and 48 h are missing due to unexpected contamination of the analyzed spheroids. Unfortunately, due to time constraints, it was impossible to perform this experiment again. The other effects influencing the presented studies are freezing and thawing procedures, which could influence the level of DNA damage. Thus, they can affect the quality of the extracted DNA and the sensitivity and specificity of the comet assay analysis.

## 5. Conclusions

Melanoma is an aggressive, often late-detected skin cancer that is challenging to treat, highlighting the need for effective therapies. One promising approach is boron neutron capture therapy (BNCT). Melanoma spheroids can serve as valuable in vitro models for testing new boron carriers and evaluating the effects of BNCT. Our study demonstrates that the uptake of BPA and cellular response to treatment can vary depending on the cell lines and culture conditions. While further investigations are needed, our findings suggest a personalized approach to selecting the appropriate treatment method, particularly in choosing boron carriers for BNCT. A deeper understanding of cell metabolism and DNA repair mechanisms, alongside the use of suitable cell models, is essential for improving the efficacy of BNCT in melanoma therapy.

## Figures and Tables

**Figure 1 cells-14-00232-f001:**
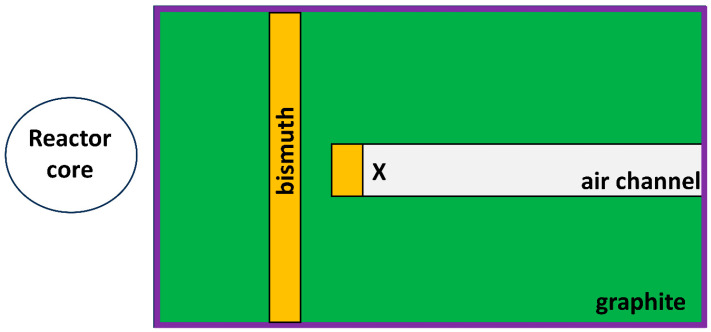
Scheme of the thermal column with the position of the cells during irradiation marked (X). Cells were irradiated at a distance of 130.95 cm from the center of the reactor core. Colors represent the walls of the thermal column made from graphite (in green), bismuth (in orange), and boral (in purple). Figure made based on Ref. [22].

**Figure 2 cells-14-00232-f002:**
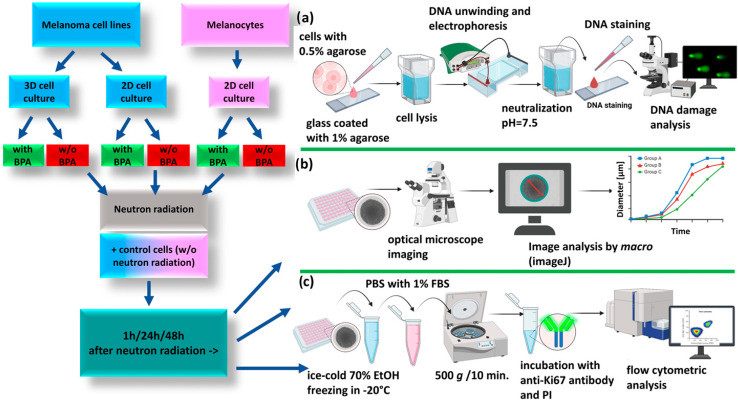
Scheme of the cell irradiation experiment. Melanoma cells (FM55p and WM266-4) and melanocytes (HEMa-LP) cultured in T25 cm^2^ flasks were irradiated with neutrons in the presence (IR_B) or absence (IR) of BPA. 7th-day melanoma spheroids formed in a 96-well low adhesive plate were irradiated with neutrons in the presence (IR_B) or absence (IR) of BPA. Control samples (non-irradiated cells, incubated (C_B), or (C) without BPA. Biological assays: (**a**) comet assay (**b**) spheroids size and shape assessment and (**c**) Ki67 protein analysis, were performed 1 h, 24 h, and 48 h after irradiation. Created with BioRender.com.

**Figure 3 cells-14-00232-f003:**
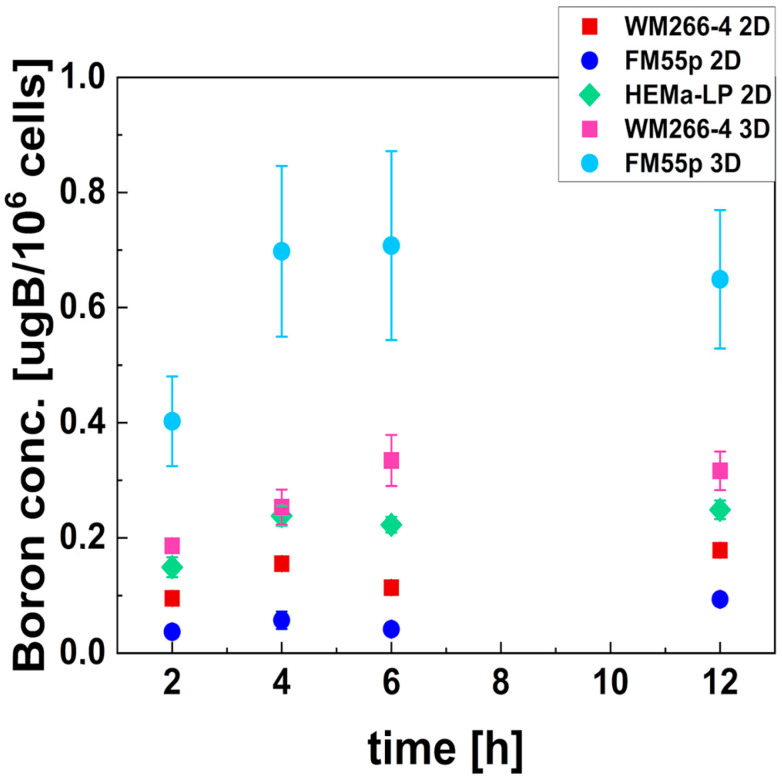
Boron concentration in cells after incubation with BPA, measured in melanoma cells (FM55p and WM266-4) and melanocytes (HEMa-LP) cultured in 2D types of culture and in the 3D model. Green diamond indicates HEMa-LP; red and pink squares—WM266-4 2D and 3D, respectively; dark and light blue circles—FM55p 2D and 3D, respectively. Data points represent mean concentration with standard deviation calculated using three measurements.

**Figure 4 cells-14-00232-f004:**
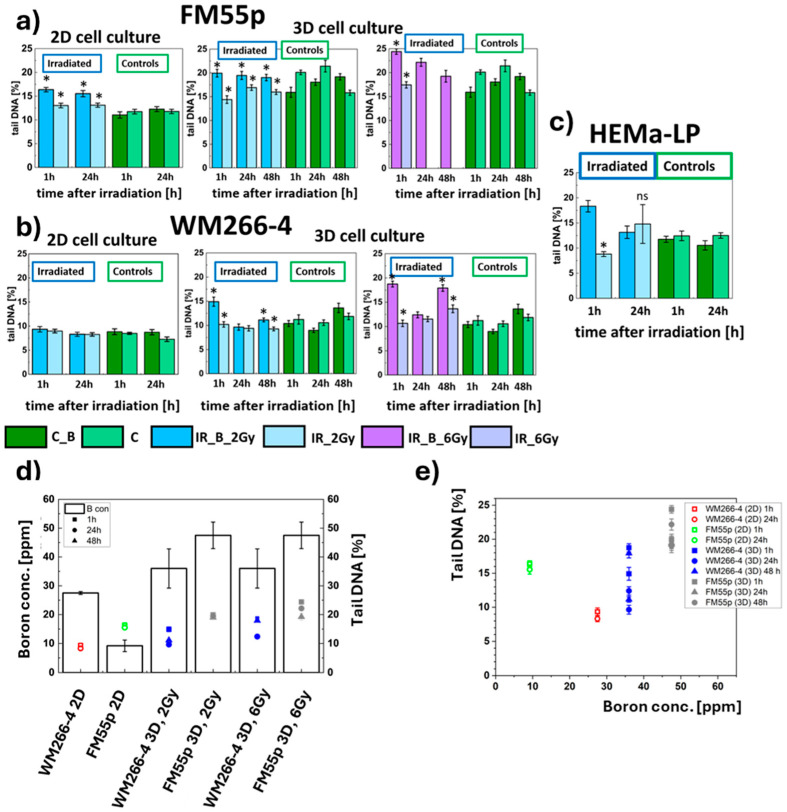
Comet assay tests results obtained for (**a**) FM55p melanoma line cells, (**b**) WM266-4 melanoma cells and (**c**) HEMa-LP melanocytes. Dark blue bars depict IR_B cells (2 Gy dose), the light blue bars indicate IR cells (2 Gy dose), the dark green bars correspond to C_B cells and light green bars indicate C cells (not irradiated). Data represent mean values with the standard deviation. * indicates statistically significant differences between IR_B and IR cells by the Student’s *t*-test (*p* < 0.05), results shown in Appendix A. The DNA damage scoring in a tail (expressed in percentage) was calculated as the DNA damage transcriptor [28]; (**d**) The ^10^B concentration (black bars) and the level of DNA damage (comet assay results) depending on cell line, type of cell culture, and delivered radiation dose. Squares, circles and triangles indicate the level of DNA damage after 1 h, 24 h, and 48 h after irradiation, respectively; (**e**) The DNA damages as a function of the ^10^B concentration for the same three points in time.

**Figure 5 cells-14-00232-f005:**
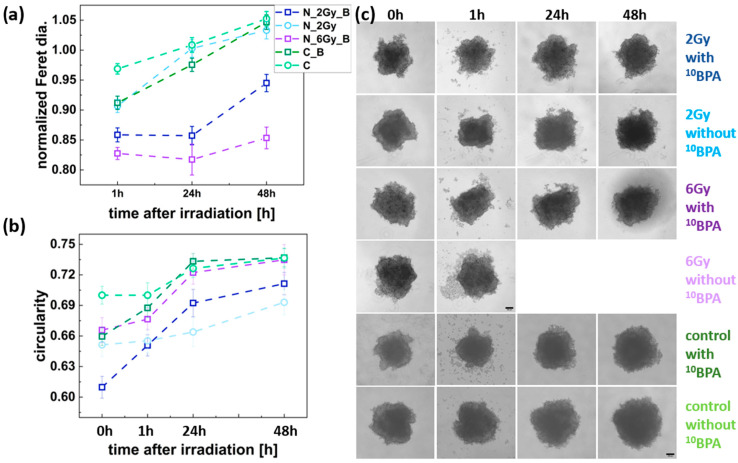
FM55p spheroid size and shape parameters. (**a**) Changes in normalized Feret diameter for the IR_B, IR, C_B, and C spheroids in time; (**b**) variations in spheroid shape (circularity) for the IR_B, IR, C_B, and C, data analyzed for n = 23–62 spheroids; (**c**) panel of representative images of the IR_B, IR, C_B, and C spheroids. Dark blue squares represent IR_B spheroids (2 Gy dose), light blue squares represent the IR spheroids (2 Gy dose); purple squares represent IR_B spheroids (6 Gy dose), dark green squares mean C_B spheroids (without neutron irradiation), and light green squares represent the C spheroids (without neutron irradiation and BPA incubation). Scale bars represent 200 µm. The estimated Feret diameters were normalized to the value obtained at 0 h.

**Figure 6 cells-14-00232-f006:**
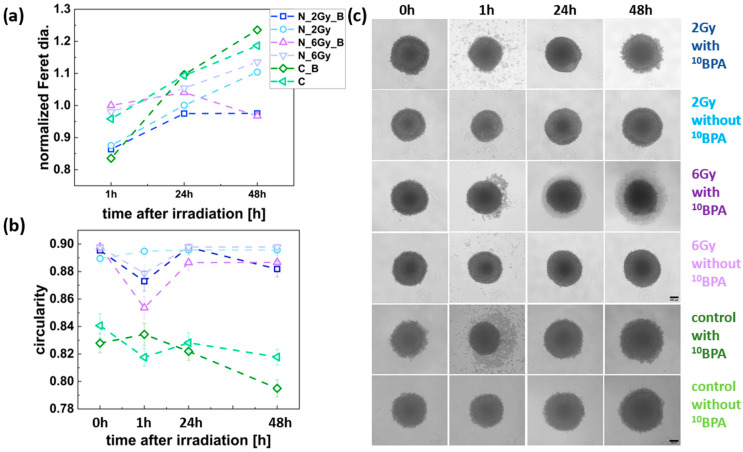
WM266-4 spheroid size and shape parameters. (**a**) Changes in normalized Feret diameter for the IR_B, IR, C_B, and C spheroids in time; (**b**) variations in spheroid shape (circularity) for the IR_B, IR, C_B, and C, data analyzed for n= 35–66 spheroids; (**c**) panel of representative images of the IR_B, IR, C_B, and C spheroids. Dark blue squares represent the IR_B spheroids (2 Gy dose), light blue squares represent the IR spheroids (2 Gy dose); purple squares represent IR_B spheroids (6 Gy dose), light purple squares correspond to the IR spheroids (6 Gy dose), dark green squares represent the C_B spheroids (without neutron irradiation), and light green squares represent C spheroids (without neutron irradiation and BPA incubation). Scale bars represent 200 µm. The estimated Feret diameters were normalized to the value obtained at 0 h.

**Figure 7 cells-14-00232-f007:**
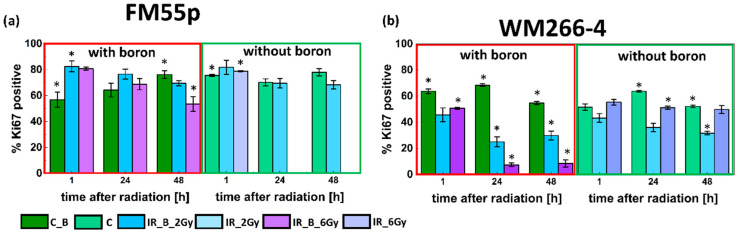
Ki67 protein expression in FM55p and WM266-4 spheroids. The graphs show the level of the Ki67 protein for spheroid-forming cells after 1 h, 24 h, and 48 h after neutron irradiation for (**a**) FM55p cell line and (**b**) WM266-4 cell line. The dark blue bar depicts the IR_B cells (2 Gy dose); the light blue bar indicates the IR cells (2 Gy dose); the dark purple indicates the IR_B cells (6 Gy dose); the light purple depicts the IR cells (6 Gy dose); the dark green bar depicts the C_B cells and light green bar indicates the C cells (not irradiated). Data represent mean values with the standard deviation. * indicates statistically significant differences in the IR_B and IR cells to the control C cells by the Student’s *t*-test (*p* < 0.05), results shown in Appendix A.

**Table 1 cells-14-00232-t001:** Operating parameters of the ICP-MS instrument.

Parameter	Value
RF power [W]	1250
Vacuum pressure [bar]	9.86 × 10^−9^
Nebulizing gas flow mL/min]	1.04
Cooling gas flow [mL/min]	17.00
Auxiliary gas flow [mL/min]	1.02
Lens voltage [V]	10

**Table 2 cells-14-00232-t002:** Boron concentrations (*C_B_*), thermal neutron fluxes (Φ_t_) and corresponding reactor powers, and doses estimated for all the irradiated cell cultures (both 2D and 3D). D_tot_ and D_0_ denote the dose delivered to the cells incubated with boron and without it, respectively. For each cell line boron-containing cells and cells without boron were irradiated with the same thermal neutron flux. Boron concentration in cells was estimated based on the studies described in Section 2.3, while the doses were calculated using Equation (1). The systematic uncertainties correspond to the unknown elemental content of the irradiated cells and correspond mainly to the assumed nitrogen mass ratio.

Cell Line	Reactor Power [kW]	Φ_t_ [cm^−2^s^−1^]	C_B_ [ppm]	D_tot_ [Gy]	D_0_ [Gy]
WM266-4 2D	20	9.60 × 10^8^	27.5 ± 0.5	2.05 ± 0.05_stat_ ± 0.14_sys_	0.33 ± 0.01_stat_ ± 0.04_sys_
FM55p 2D	60	2.88 × 10^9^	9.2 ± 2.0	2.71 ± 0.38_stat_ ± 0.40_sys_	0.98 ± 0.02_stat_ ± 0.04_sys_
HEMa-LP	20	9.60 × 10^8^	26.4 ± 1.7	1.98 ± 0.12_stat_ ± 0.14_sys_	0.33 ± 0.01_stat_ ± 0.04_sys_
WM266-4 3D	17	8.16 × 10^8^	36.0 ± 6.8	3.14 ± 0.48_stat_ ± 0.07_sys_	0.61 ± 0.03_stat_ ± 0.07_sys_
FM55p 3D	12	5.76 × 10^8^	47.5 ± 4.6	2.78 ± 0.24_stat_ ± 0.05_sys_	0.43 ± 0.02_stat_ ± 0.03_sys_
WM266-4 3D	51	2.45 × 10^9^	36.0 ± 6.8	9.33 ± 1.50_stat_ ± 0.2_sys_	1.82 ± 0.09_stat_ ± 0.20_sys_
FM55p 3D	36	1.73 × 10^9^	47.5 ± 4.6	8.30 ± 0.80_stat_ ± 0.14_sys_	1.28 ± 0.07_stat_ ± 0.14_sys_

**Table 3 cells-14-00232-t003:** Values of the dose rates due to background processes ($R_{bg}$) and the ^10^B (n,α)^7^Li reaction for both the 2D and 3D cell cultures estimated by Monte Carlo simulations assuming the reactor power $P_{max}$ = 250 kW and boron concentration of 1 ppm. The contributions due to the neutron scattering on hydrogen ($R_{H}$), the ^14^N (n,p)^14^C reaction ($R_{N}$), and γ-rays ($R_{\gamma }$) are also reported.

Cell Culture	$R_{bg}$ [mGy/s]	$R_{B}$ [mGy/s]	$R_{H}$ [mGy/s]	$R_{N}$ [mGy/s]	$R_{\gamma }$ [mGy/s]
2D	9.79 ± 0.09	1.31 ± 0.01	0.699 ± 0.001	3.28 ± 0.01	5.81 ± 0.07
Spheroids	14.98 ± 0.08	1.70 ± 0.03	0.119 ± 0.005	4.26 ± 0.08	10.6 ± 0.7

## Data Availability

The raw data supporting the conclusions of this article will be made available by the authors on request.

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
