# Peer review of "Effect of Neutron Radiation on 10BPA-Loaded Melanoma Spheroids and Melanocytes"

_cells, 2025, doi:10.3390/cells14030232_

Round 1

Reviewer 1 Report

Comments and Suggestions for Authors

The authors in this article study the difference in DNA damage induced after BNCT treatment between 2D and 3D cell culture models, focusing on the fact that 3D cultures in the form of spheroids show greater similarity to the tumour and tumour microenvironment such that the observed response to BNCT is more physiological than that observed in 2D cultures.  Furthermore, they also indicate the difference in boron uptake between the two types of cell cultures. They conclude that spheroids can be used to assess boron transporters and the effects of BNCT more efficiently, although they say that it all depends on the cell model being studied.

The article is very interesting and opens up new ways to determine the dose in this type of therapy more effectively and thus enhance its efficacy while minimising possible side effects. However, the authors should clarify some aspects that are not clear in the manuscript:

-In point 2.5, line 155, the authors calculate the doce using the elemental composition of cells derived from average adult male healthy soft tissue may not provide sufficient accuracy for dose calculations. In neutron dose calculations, particularly for thermal neutrons, it is crucial to account for the nitrogen content specific to each cell line, as this significantly affects the physical dose. The impact of nitrogen content on dose calculation and a technique for measuring cellular nitrogen percentage are discussed in detail en diferentes publicaciones. Therefore, I recommend measuring the nitrogen content using the aforementioned technique. If this is not feasible, it is essential to explicitly acknowledge the simplifications made and discuss their potential impact on the accuracy of the dose calculation.

Moreover, the experiments are carried out on the culture flask and on plates in the case of spheroids. Do the T75 flasks used in the 2D cultures contain borosilicates? The authors do not indicate this, because if this compound is present, the results obtained must take into account the effect of the neutron with this boron in the flask. Please explain this section.

Figure 2 is not clearly visible, the image definition needs to be improved.

-Point 2.6, line 252. The DNA damage experiment using the comet assay, the authors freeze the cells before performing the comet assay. The results obtained can be affected by freezing and thawing, as these processes can themselves induce DNA damage by the formation of crystals, leading to breaks in the strand that could be misinterpreted as genotoxic damage, and the thawing process can increase the production of reactive oxygen species (ROS), which also damage DNA. All this can affect the quality of the extracted DNA, affecting the sensitivity and specificity of the analysis.

Why has the study not been carried out with unfrozen cells, using time kinetics after irradiation? The authors should justify this. I advise the authors to perform an experiment with non-frozen cells to support the results obtained with frozen cells.

-Point 2.3 line 301. Boron uptake in primary melanocytes has only been studied in 2d cultures, why has it not been carried out in 3d cultures as in the other lines? This result is important to complete this study and to know the differences in an effective way. Why did they not use the 6Gy dose in the HEMa-LP line? The study should be completed with these experiments.

-To clarify the results and to see if the difference in response between the two tumour lines is due to boron uptake, the authors should provide a graph relating boron uptake to DNA damage induced after irradiation.

Figure 4a. Non-boron controls in 3d cultures at 24 hours show almost the same damage as those irradiated with boron. Furthermore, the boron controls show less DNA damage than the non-boron controls, does boron positively affect cell viability? The authors should justify this result.

-Line 373. There is a part in this sentence that is in Polish.

- The graph in figure 5 b shows how the addition of boron alone induces loss of circularity in spheroids.  The authors should explain this result. Also, why is the 6Gy result without boron missing?

- Figure 7a. The data for FM55p irradiated at 6Gy without boron at 24 and 48 hours are missing.  In addition, the graphs should have better resolution.

Reviewer 2 Report

Comments and Suggestions for Authors

In this work, the authors constructed Melanoma spheroids as in vitro models for testing BNCT. They also analyzed boron uptake, DNA damage, cell growth, and proliferation using BPA as a boron carrier, providing a valuable research method for situations where in vivo mouse tumor experiments are not feasible.

However, there are several issues that require further clarification:

Please explain why spheroids have higher boron uptake compared to monolayer cells of the same type. Is it because boron in the intercellular spaces is not washed away, thus affecting detection, or is it due to higher expression levels of LAT-1 in the spheroid state?

In the comet assay, there are some doubts about the experimental results: 1) Why is there about 10% tail DNA in the control group? 2) Why does 2 Gy of irradiation not cause DNA damage in the 2D culture group of the WM266-4 cell line? 3) Representative comet images need to be displayed.

In Figures 5a, 5b, 6a, and 6b, the different starting values of the data will affect the rigor of the experimental results.

What is the relationship between spheroid size and shape parameters and the efficacy of BNCT? The fact that circularity shows little change or even increases 24 or 48 hours after treatment, does this indicate that BNCT has not caused cell damage, and is this in line with expectations?

In Figure 7, explain the reasons for the different results between the two cell lines.

Additionally, there are some minor questions:

For Figure 4, please clarify the figure legend to indicate the corresponding content for each panel. The current legend does not include the figure numbers.

Reviewer 3 Report

Comments and Suggestions for Authors

This study aims to advance our knowledge of the best cellular models for studying the efficacy of BNCT in vitro. To this end, the authors analyze DNA damage and proliferation in response to neutron irradiation in 2D and 3D cultures of BPA-treated melanoma cell lines. The manuscript provides useful information. However, some points should be addressed for publication:

Specific Comments:

1. The Materials and Methods section should include a description of the statistical analysis performed.

2. How has the composition of the culture medium been taken into account in the dose calculation? The reference 22, to which authors allude, does not seem to include this data.

3. The authors propose, for simplicity, to refer to high doses as 6 Gy doses. However, this is inaccurate and may mislead subsequent studies that may be based on these data, since high doses range from 8.3-9.4 Gy.

4. The authors study "melanoma cells in two models (2D and 3D cell cultures) as well as melanocytes as a control (2D) to assess DNA damage after BNCT and neutron irradiation". However, they do not perform the same studies in all cell lines and conditions. To highlight the differences between the different culture conditions they should: a) Irradiate also at high doses the melanocyte line and the 2D melanoma cultures. b) Do the comet assay at 48h in all cases. c) Perform proliferation assays on 2D cultures of melanoma and melanocytes.

5.  Why are the boron concentration data in Figure 3 shown in “g/million cells” if “ppm” are used for the calculation of the dose received in the irradiation experiments (Table 2)? Authors should represent data in figure 3 also in “ppm” or indicate the equivalence, in order to understand where the concentration values in table 2 come from. In addition, in figure 3 they should indicate the number of experiments performed to obtain the values represented.

6. In figure 4a, the data of FM55p cells irradiated at 6Gy without BPA at 24 and 48h are missing. Anyway, controls (non-irradiated) cells show similar levels of damage to the samples irradiated at 6 Gy (with or without BPA) and even higher than some of those irradiated at 2Gy. Something similar is also observed for some conditions in figure 4b. Authors should explain the comparison in all conditions, with all controls. In appendix B they include the statistical significance, but in some cases significant differences are due to a higher percentage of damage in the controls than in the irradiated samples, so, with these results, they cannot conclude that there is DNA damage in response to irradiation. All this needs to be further clarified and discussed.

7. The statistical significance shown in figure 7 is not clear. Are irradiations with boron compared to irradiations without boron? Then, why in some cases is there an asterisk in equivalent bars (with boron and without boron) and in others there is no asterisk? This should be better indicated either in the graph or in the figure legend.

8. The discussion section states that “boron concentration in melanoma is dependent on the duration of incubation with BPA, reaching its maximum after 4 hours” ((lines 433-434). However, in 3D cultures, it seems that boron accumulation is higher at 6 h than at 4 h. It would be interesting to study an intermediate point between 6 and 12 hours.

9. The discussion of cell cycle arrest in G2/M phase or G0 phase from the results obtained with the proliferation marker Ki67 is complex. Analysis of cell cycle and apoptosis induction would help to complete and better discuss these results.

10. According to what has been explained in points 4 and 6, the following sentences of the abstract are not proved and should be modified:

-   “Results indicated that IR_B spheroids had … and significantly higher DNA damage compared to other groups”.

-    “Melanoma cells in the 3D model showed higher DNA damage levels than those in the 2D model”.

 Minor points:

-   Figure 6a is not referenced in the text.

- In the figure legends 4 and 7 the sentence ‘Data represent mean values with standard deviation’ is repeated.

- The value of the spheroid diameter at time 0h is missing in figures 5a and 6a although this time is shown in the images (fig. 5c and 6c) and in figures 5b and 6b. It should also be included in figures 5a and 6a.

Round 2

Reviewer 1 Report

Comments and Suggestions for Authors

The authors have corrected and responded satisfactorily to my suggestions. It is true that some of the questions I suggested regarding DNA damage remain to be explained, although I understand that the experimentation and re-doing of the experiments is limited as they depend on neutron beam time. As such, and knowing this limitation, I accept the publication. 

Author Response

We appreciate very much the preparation of the second review report by Reviewer 2, and we are pleased that Reviewer 2 accepted the manuscript for publication. 

Reviewer 2 Report

Comments and Suggestions for Authors

The author has made modifications according to the reviewer's comments.

Author Response

We appreciate very much the preparation of the second review report by the Referee 1, and we are pleased that Reviewer 1 accepted the manuscript for publication. 

Reviewer 3 Report

Comments and Suggestions for Authors

The authors have answered to most of the suggestions and questions and the manuscript has been improved. However, one important aspect concerning the comet assay remains to be resolved and addressed. It is understood that it is difficult to perform some experiments in the reactor due to time and space constraints, but this should not affect the publication of robust results. It is not possible to conclude that ‘IR_B spheroids might show more DNA damage’ or that ‘they are more susceptible to BNCT than 2D cell cultures’, while justifying that the high percentage of DNA damage observed in control spheroids (in some cases equal or higher than in IR_B) is due to the freeze-thaw process of the samples. The conclusion does not hold.

Author Response

We appreciate very much the preparation of the second review. 

We have prepared a new version of the manuscript taking the Referees’ comments into account. We have modified the following text in Abstract (lines 38 – 41):

“Results indicated that IR_B spheroids might exhibit reduced diameters and higher DNA damage than other groups. Melanoma cells in the 3D model showed that their DNA damage levels may be higher than those in the 2D model.”

to:

„The results indicated that IR_B spheroids might exhibit a reduced diameter. Melanoma cells in the 3D model showed that their DNA damage levels may be higher than those in the 2D model.”;

In the Discussion section (lines 536-539) the text:

“The results of the comet assay may suggest that spheroids are more susceptible to BNCT than 2D cell cultures. Similar findings were published by Yu et al., revealing higher DNA damage in spheroids of human pancreatic cancer cell lines than in 2D culture [43].”;

was modified to:

“On the whole, the results of the comet assay might suggest that spheroids are more susceptible to BNCT than 2D cell cultures. This is in line with the findings published by Yu et al. [43], revealing a higher DNA damage in spheroids than in 2D culture for human pancreatic cancer cell lines. However, the variability of the results obtained by the comet assay needs to be further investigated. To this purpose, additional experiments will be carried out to clarify the variability observed and to get to a robust conclusion.”

We hope that the modified version of the manuscript is acceptable for publication.